# Post-Treatment Occurrence of Serum Cryoglobulinemia in Chronic Hepatitis C Patients

**DOI:** 10.3390/diagnostics14111188

**Published:** 2024-06-05

**Authors:** Gantogtokh Dashjamts, Amin-Erdene Ganzorig, Yumchinsuren Tsedendorj, Ganchimeg Dondov, Otgongerel Nergui, Tegshjargal Badamjav, Chung-Feng Huang, Po-Cheng Liang, Tulgaa Lonjid, Batbold Batsaikhan, Chia-Yen Dai

**Affiliations:** 1Department of Internal Medicine, Institute of Medical Sciences, Mongolian National University of Medical Sciences, Ulaanbaatar 14210, Mongolia; gantogtokh.ims@mnums.edu.mn (G.D.); aminerdene.ims@mnums.edu.mn (A.-E.G.); yumchinsuren.ims@mnums.edu.mn (Y.T.); ganchimeg.ims@mnums.edu.mn (G.D.); otgongerel.ims@mnums.edu.mn (O.N.); tegshjargal.ims@mnums.edu.mn (T.B.); tulgaa.ims@mnums.edu.mn (T.L.); 2Department of Biological Sciences, School of Life Sciences, Inner Mongolia University, Hohhot 010031, China; 3Department of Internal Medicine, Kaohsiung Medical University Hospital, Kaohsiung Medical University, Kaohsiung 807378, Taiwan; fengcheerup@gmail.com (C.-F.H.); pocheng.liang@gmail.com (P.-C.L.); 4Department of Occupational and Environmental Medicine, Kaohsiung Medical University Hospital, Kaohsiung Medical University, Kaohsiung 807378, Taiwan; 5Department of Health Research, Graduate School, Mongolian National University of Medical Sciences, Ulaanbaatar 14210, Mongolia; 6Ph.D. Program in Environmental and Occupational Medicine, Drug Development and Value Creation Research Center, Kaohsiung Medical University, Kaohsiung 807378, Taiwan; 7College of Professional Studies, National Pingtung University of Science and Technology, Pingtung 91201, Taiwan

**Keywords:** cryoglobulinemia, cryoprecipitate, occurrence, hepatitis C virus, antiviral therapy

## Abstract

Background: Persistent cryoglobulinemia after the completion of antiviral treatment is an important consideration of clinical management in chronic hepatitis C patients. We aimed to investigate the occurrence of serum cryoglobulinemia in chronic hepatitis C patients without cryoglobulinemia at the initiation of antiviral treatment. Methods: In total, 776 patients without cryoglobulinemia were assessed for serum cryoglobulinemia after the completion of anti-HCV treatment. Serum cryoglobulinemia precipitation was assessed upon both the initiation and the completion of the treatment and analyzed for the clinical laboratory factors associated with chronic hepatitis C. Results: One hundred eighteen (118) patients were checked for serum cryo-precipitation after the completion of the treatment, and eight patients (4.6%) were positive for serum cryoglobulinemia. The patients who tested positive for cryoglobulinemia included a higher proportion of liver cirrhosis patients (4/50%, *p* = 0.033) and other organ cancer patients (5/62.5%, *p* = 0.006) than patients who showed no signs of cryoglobulinemia after treatment. In a multivariate analysis, liver cirrhosis (odds ratio [OR]—17.86, 95% confidence interval [95% CI]—1.79–177.35, *p* = 0.014) and other organ cancer (OR–25.17 95% CI—2.59–244.23, *p* = 0.005) were independently and significantly associated with positive cryoglobulinemia 3 months after antiviral treatment. Conclusions: Three months after the antiviral DAA therapy had concluded, eight patients tested positive for cryoglobulinemia, representing a 6.7% prevalence. Liver cirrhosis and other organ cancer were independently and significantly associated with positive cryoglobulinemia after antiviral treatment. Further investigation into the causes of positive cryoglobulinemia after DAA antiviral therapy is warranted.

## 1. Introduction

Hepatitis C virus (HCV) infection is highly prevalent worldwide and is known to be associated with extrahepatic manifestations. One of the most commonly reported clinical extrahepatic symptoms is mixed cryoglobulinemia (MC) [1]. The clinical manifestation of MC includes arthralgia, fatigue, and/or myalgia, as well as small and medium vessel vasculitis. The cause of MC in HCV-infected patients is considered to be B-cell clonal expansion. Circulating serum cryoglobulinemia can be detected in 40 to 60% of HCV-infected patients, and clinical vasculitis is observed in less than 10% of these cases [2]. Despite ongoing HCV or the presence of serum anti-HCV antibodies occurring in 90% of cases with MC and a high concentration of anti-HCV antibodies, HCV RNA has been detected in the cryoprecipitate of patients’ serum [3,4], suggesting the crucial role of viral particles in cryoglobulinemia development [5,6]. Risk factors such as older age, longer duration of HCV infection, and kidney involvement are associated with approximately 80% of patients with cryoglobulinemic vasculitis and HCV infection [7,8]. Cryoglobulinemia is also linked with advanced liver fibrosis and liver steatosis in patients with chronic hepatitis C (CHC) [9,10]. Although CHC patients with MC, MC vasculitis, and chronic kidney manifestations are carefully monitored during interferon-based or interferon-free antiviral treatment, further interventions may be required based on their progress [11].

MC is characterized by cryoglobulins (temperature-related immunoglobulins), that precipitate in serum at low temperatures and dissolve reversibly at 37 °C. It is an immune complex (mostly mono- or polyclonal immunoglobulin M) that includes immunoglobulins, small particles of HCV, cytokines, rheumatoid factor, and C protein [12]. Therefore, MC may not always be associated with clinical symptoms, or it may be linked to more severe manifestations such as systemic vasculitis with or without renal and neurological symptoms [13].

The clinical remission of MC syndrome in chronic HCV infection is often associated with the clearance of the virus [14]. Some studies conducted in Europe and America have shown that antiviral treatment against HCV, including interferon-based therapies [15] and direct antiviral agents [16], are effective in treating MC associated with HCV. MC has been reported to remain stable after achieving sustained virological response (SVR); however, it is considered a rare disease in Asia [17]. The effectiveness of anti-HCV therapy in treating MC associated with HCV in Asians remains controversial.

Treatment for HCV patients with cryoglobulinemia is recommended when there is a progressive organ-threatening illness. Recent guidelines recommend antiviral treatment alone for CHC patients without any symptoms of cryoglobulinemia. In the majority of instances, the effective management of HCV results in the amelioration of cryoglobulinemia symptoms and complications [18,19]. Nevertheless, MC is not correlated with relapse during antiviral treatment, and the persistence of cryoglobulinemia in patients is linked to more complex ailments [20]. Due to cost-effectiveness, the standard treatments for HCV-infected patients in most Asian countries [21] where direct-acting antiviral agents (DAAs) are not available continue to be peg-interferon and ribavirin. However, interferon-free DAA regimens have successfully resolved 50% of cryoglobulinemia cases, with a high response rate reported [22,23]. Also, there are a number of patients who still experience MC after completion of antiviral treatment, which is related to advanced liver fibrosis and more complicated clinical manifestations [24].

Antiviral therapy’s clearance of HCV is suggested to reduce the clinical manifestations of cryoglobulinemia and decrease cryoglobulin production [25]. However, persistent cryoglobulinemia symptoms may arise even after achieving SVR through pegylated interferon alpha plus ribavirin treatment [26]. It is suggested to administer high doses and prolong the duration of antiviral treatment for patients with cryoglobulinemia [27]. Given that Taiwan and Mongolia are endemic areas for HCV infection, where antiviral treatment yields a higher SVR rate [28,29,30], our research team undertook this study to explore the potential for cryoglobulinemia-negative patients to become positive after antiviral treatment and to examine the factors associated with cryoglobulinemia positivity in patients.

## 2. Materials and Methods

### 2.1. Patients

We conducted a comprehensive study that included patients with CHC admitted to Kaohsiung Medical University Hospital between 2005 and 2016. Initially, a total of 1306 patients underwent screening for cryoglobulinemia at the commencement of antiviral treatment. Among them, 776 patients exhibited no signs of cryoglobulinemia. Subsequently, 118 patients who underwent cryoglobulinemia screening after completing antiviral therapy were included in this study population for further analysis of associated factors. These patients were randomly selected from cryoglobulinemia-negative patients at the initiation of antiviral treatment and also checked for cryoglobulinemia after 3 months of completion of antiviral treatment. The scheme chart illustrating the population is presented in Figure 1.

CHC diagnosis was based on the detection of serum HCV RNA, assessed using the COBAS AMPLICOR Hepatitis C Virus Test, version 2.0 (Roche, Branchburg, NJ, USA), with a detection limit of 50 IU/mL. Additionally, all patients underwent biochemical assays to evaluate liver function, fat content, and iron levels. Exclusion criteria encompassed patients with (a) concurrent connective tissue disorders; (b) prior treatment for HCV infection and/or exposure to immunosuppressive drugs; (c) historical or current alcohol consumption; (d) evidence indicating other liver pathologies such as autoimmune hepatitis, primary biliary cirrhosis, or hepatocellular carcinoma; (e) undiagnosed cryoglobulinemia precipitates; (f) pregnancy; (g) physical or mental incapacity to participate in the study; (h) absence of diagnostic and laboratory parameters.

We also checked for HBV coinfection, HIV coinfection, decompensated cirrhosis, HCC, diabetes, hypertension, hyperlipidemia, cerebrovascular events, cardiovascular events, chronic kidney disease, and cancer in other organs as possible risk factors associated with cryoglobulinemia after achieving SVR by DAA treatment.

### 2.2. Treatment Regimen

All patients were treated with direct-acting antiviral drugs following the guidelines recommended by the APASL. Treatment considerations included factors such as HCV genotype, viral loads, and viral response, with a standard treatment duration of 12–24 weeks as stipulated by the reimbursement guidelines of the Taiwan National Health Insurance [31]. The study’s design, compliant with ethical standards, received approval from the Ethics Committee of Kaohsiung Medical University Hospital, ensuring the protection of participants’ rights. All clinical investigations strictly adhered to the principles outlined in the Declaration of Helsinki, guaranteeing the ethical conduct of research. This standardized approach to treatment and research methodology aims to uphold the highest standards of patient care and scientific integrity in the pursuit of understanding and managing chronic hepatitis C effectively. 

### 2.3. Laboratory Tests

#### 2.3.1. Detection of Cryoglobulinemia

Serum cryoprecipitation was conducted using previously published method [32,33]. Briefly, 10–15 mL of blood was incubated in a water bath at 37 °C. Subsequently, up to 5 mL of serum was separated by centrifugation and stored under refrigeration (4 °C). Serum was then observed for cryoprecipitation once daily for 7 consecutive days. If cryoprecipitation was observed, the tubes were re-incubated at 37 °C for 30 min to verify dissolution of the cryoprecipitate (Figure 2). We confirmed the diagnosis of cryoglobulinemia by observing the presence of cryoprecipitate in the serum of CHC-proven patients under refrigeration, as well as its dissolution at 37 °C, both at the beginning and end of the treatment.

Prior to treatment initiation, we collected general demographic characteristics and conducted serum biochemical analyses using commercial tests. These parameters included aspartate aminotransferase (AST), alanine aminotransferase (ALT), alpha-fetoprotein (AFP), and platelet count. HCV genotypes were determined using amplicons generated by the Amplicor HCV test and a commercially available assay (Line Probe assay, LIPA HCV, Innogenetics, Gent, Belgium). The HCV genotypes were classified according to the method proposed by Abbott RealTime HCV Genotype II, Abbott Molecular, Des Plaines IL, USA [34]. 

SVR was defined as negative HCV RNA at 6 months after cessation of treatment. Advanced fibrosis was assessed using the Fibrosis-4 (FIB4) index [35], which was calculated using the following formula:FIB4 = (age [years] × AST [U/L])/(Platelet [10^9^/L] × √ALT [U/L])

Aspartate aminotransferase to Platelet Ratio Index (APRI) was calculated using formula [35]:
APRI = (GOT [U/L]/GOT [Upper limit of normal range])/Platelet [10^9^/L]

Child–Turcotte–Pugh score was evaluated by original method [36,37].

#### 2.3.2. Histological Evaluation

Liver biopsies were obtained from the patients before antiviral therapy and assessed using the scoring system described by Knodell and Scheuer [38,39]. The histological evaluation was performed by one pathologist experienced in hepatopathology, who was blinded to the clinical data. 

### 2.4. Statistical Analysis

We computed mean values and standard deviations for continuous variables. Continuous and categorical variables were compared using Student’s *t*-test, Chi-square (X^2^), or Fisher’s exact test, as applicable. To evaluate the relationship between associated factors and persistent cryoglobulinemia, we conducted both univariate analysis and multivariate logistic regression analysis. Patient data were originally collected and organized using Microsoft Excel software (version 2016). Statistical significance was defined at a *p*-value below 0.05, considering two-sided hypotheses. All statistical computations were performed using IBM SPSS Statistics for Windows, Version 20.0, developed by IBM Corp. in Armonk, NY, USA.

## 3. Results

Out of the 1306 total patients initially screened for cryoglobulinemia at the onset of antiviral treatment, 776 patients showed no signs of cryoglobulinemia. Subsequently, we conducted a follow-up screening for cryoglobulinemia 3 months after the completion of treatment in a subset of these patients. Specifically, 118 patients underwent this follow-up assessment, revealing that 8 patients tested positive for cryoprecipitation in serum, while the remaining patients tested negative (Figure 1).

The demographic characteristics of the 118 patients who were available for serum cryoglobulinemia detection 3 months after antiviral therapy are presented in Table 1. Eight patients (6.7%) exhibited cryoglobulinemia positivity after treatment. Individuals with cryoglobulinemia were relatively older than those without cryoglobulinemia (72.1 ± 10.4; 64.2 ± 11.4, *p* < 0.001). Other factors such as gender, body mass index, HCV genotype, and other HCV-related diseases did not show significant associations with cryoglobulinemia positivity. Among the laboratory parameters, low white blood cells (4.7 ± 1.3/SD, *p* = 0.001) and low creatinine levels (1.3 ± 0.3, *p* = 0.012) were significantly associated with cryoglobulinemia positivity. 

Compared with patients without cryoglobulinemia, the patients with positive cryoglobulinemia were more frequently male. They also had higher FIB4 indexes, APRI scores, ALT and AST levels, and AFP rates. Additionally, they exhibited lower HCV RNA, triglyceride levels, and platelet counts (Table 1).

One patient had an HBV infection, and none had HIV infections among the patients with cryoglobulinemia. The patients who tested positive for cryoglobulinemia included a higher proportion of liver cirrhosis patients (4/50%, *p* = 0.033) and other organ cancer patients (5/62.5%, *p* = 0.006) compared to patients who showed no sign of cryoglobulinemia after treatment. Compared to individuals without cryoglobulinemia, those with cryoglobulinemia had no difference in the characteristics of comorbidities such as HCC, diabetes, hypertension, hyperlipidemia, chronic kidney disease, cardiovascular disease, and cerebrovascular disease (Table 2).

In multivariate logistic regression analysis, liver cirrhosis (odds ratio [OR]—17.86, 95% confidence interval [95% CI]—1.79–177.35, *p* = 0.014) and other organ cancer (OR-25.17 95% CI—2.59–244.23, *p* = 0.005) were found to be independently and significantly associated with positive cryoglobulinemia 3 months after antiviral treatment (Table 3).

## 4. Discussion

In this cohort study, we investigated the presence of cryoglobulinemia following completion of antiviral therapy. When 118 patients were screened for cryoglobulinemia 3 months after completing antiviral therapy, 8 patients tested positive, resulting in a prevalence of 6.7%. Our findings suggest that CHC patients may develop cryoglobulinemia after antiviral therapy, and this occurrence was associated with older age, liver cirrhosis, and cancer in other organs. However, Younossi Z et al. reported in a meta-analysis that the pooled prevalence of asymptomatic MC was 30.1% in HCV-infected individuals and 1.9% in non-HCV populations. The prevalence of symptomatic MC was 4.9% in the HCV-infected population [26]. In our study, we investigated cryoprecipitation because these patients may have a higher risk of developing severe vasculitis, skin, or neurologic symptoms. However, none of the 118 patients had immunosuppressive treatment during antiviral treatment because these patients did not have any sign of vasculitis or purpura.

Our previous findings indicated that the prevalence of cryoglobulinemia ranged from 30.4–32% in Asian CHC patients. Serum cryoglobulinemic precipitation was strongly associated with advanced fibrosis, whether liver fibrosis was assessed by liver biopsy or the FIB4 biomarker. Cryoglobulinemia was an independent factor in the development of advanced liver fibrosis, and the actual reason for this association remains uncertain [10]. On the other hand, we have studied that 34.5% of patients who completed antiviral treatment can test positive for serum cryoglobulinemia, and it is related to advanced fibrosis [24]. In both studies, we investigated serum cryoprecipitation because this condition may increase the risk of developing severe vasculitis, skin, or neurologic symptoms in CHC patients. The results of the recent study have revealed the possibility of positive cryoglobulinemia after achieving SVR by DAA treatment and its independent association with liver cirrhosis and cancer in other organs. Taken together, it can be concluded that serum cryoglobulinemia has a strong relationship with advanced liver fibrosis and liver cirrhosis at the initiation of antiviral treatment, after completion of the treatment, and after successful treatment in CHC patients. It is clear that the medical state (liver function or the total number of normal hepatic cells) is a crucial factor in the development of cryoprecipitation in CHC patients, or the development of cryoprecipitation (immune complex) may worsen the medical state of CHC patients.

Stasi C et al. have investigated the association between B-cell depletion and liver stiffness following a combination of antiviral therapy and rituximab [40]. Gragnani L et al. noted that achieving viral eradication led to a decrease in MC syndromes, and persistent clinical manifestations of MC post-antiviral therapy were uncommon [15]. Their prior research also indicated that identified cryoglobulinemia was an independent prognostic factor for not achieving SVR by pegylated interferon plus ribavirin therapy [15]. Moreover, numerous research studies have investigated the therapeutic efficacy of pegylated interferon treatment, consistently demonstrating that HCV-infected patients with cryoglobulinemia exhibited lower rates of SVR compared to cryoglobulinemia-negative patients [41,42,43]. Additionally, patients with CHC and MC can undergo treatment with standard interferon-based antiviral therapy combined with rituximab [44]. Cryoglobulinemia was independently associated with liver cirrhosis by biopsy and FIB4 index in multivariate analysis [10]. Saadoun et al. explored the correlation between HCV-infected cryoglobulinemia patients and a heightened propensity for liver fibrosis and steatosis in patients with confirmed biopsy results [9]. The responsible mechanisms linking MC and cirrhosis remain unclear. However, studies have reported that HCV-infected patients show overexpression of memory B-cells, with memory B-cell CD86 levels being associated with advanced liver fibrosis in CHC [45]. Furthermore, during HCV infection, activated memory B-cells have been associated with a heightened frequency of T helper 1 cells [46]. Patients with HCV-induced MC exhibit an association between disrupted B-cell homeostasis due to naïve B-cell apoptosis [47].

Advanced fibrosis (F3 and F4) was described by a cut-off of 3.25 in the FIB4 score in non-biopsy patients, which includes age, AST, ALT, and platelet levels. Advanced age, a factor previously linked to cryoglobulinemia patients, as well as lower platelet levels, were identified as associated factors in CHC patients with cryoglobulinemia. Taken together, the FIB4 index serves as a simple, non-invasive marker to predict liver fibrosis in CHC due to its strong correlation with liver biopsy. In our study, although the liver fibrosis score was higher in patients with cryoglobulinemia, it did not reach statistical significance. Some other studies have reported that patients with liver steatosis [9] and fibrosis have an impact from cryoglobulinemia-related manifestations [9,48]. It is unclear what the underlying reasons for the impact of cryoglobulinemia on liver fibrosis are. Nevertheless, previously published studies have suggested that the impact of cryoglobulinemia on fibrosis could be due to certain symptoms and clinical manifestations of the disease [49] or its relationship with the duration of progressing disease [50]. Another report attributed the association between cryoglobulinemia and fibrosis to the inclusion of patients with alcohol consumption in the study [51]. Also, there is a link suggesting that cryoglobulinemia is associated with liver cell necroinflammation [52]. Hence, it is apparent that the fundamental pathological association between liver fibrogenesis and cryoglobulinemia is still under consideration. An acceptable explanation for this association is that a decrease in liver blood perfusion may result in changes in liver Kupffer cells, which could, in turn, reduce the clearance of the circulating immune complex, including cryoglobulins [4]. 

The positive correlation between IgG and IgM levels in MC is due to the expansion of B-cells producing both IgG and IgM [53]. Negative triglyceride and very low-density lipoprotein association have been reported in HCV-infected patients, and it is consistent in MC patients [54]. Further investigation of low lipid profiles in MC patients is necessary.

Some patients with SVR have persistent or long-term mixed cryoglobulinemia [55]. However, 6.7% of our SVR patients still had MC after completion of antiviral treatment. MC is a common HCV-induced autoimmune disease [53] involving a complex interplay between environmental factors, genetic susceptibility, immunodeficiency, and hormones [56]. The persistence of MC after viral clearance suggests that HCV antigens are not components of cryoprecipitated immune complexes and emphasizes the importance of genetic studies of long-term MC in patients with SVR. Additionally, chronic HCV infection significantly disrupts the B-cell compartment and generates multiple large B-cell clones [57]. Therefore, persistent or long-term MC after SVR may indicate that some B-cells have reached the “irreversible” stage of generating pathogenic clones before viral clearance [58].

A lower rate of cryoglobulinemia prevalence was detected in the patients with HCV/HBV and HCV/HIV co-infection in this study. It seems that HBV infection and HIV infection do not appear to play a significant role in cryoglobulin production. Patients with HCV infections are known to have low triglyceride levels [59], and no association was found in our study. Additionally, HCV genotype may have an impact on the composition of cryoglobulin products. Further analysis is warranted to delve into the molecular mechanisms underlying the presence of cryoglobulinemia across different HCV genotypes. Understanding these mechanisms will not only deepen our comprehension of the disease process but also pave the way for targeted therapeutic interventions. By elucidating the molecular pathways involved, we can develop more effective treatment strategies tailored to specific genotypes. This approach has the potential to significantly improve patient outcomes in the management of cryoglobulinemia associated with HCV infection, particularly in patients with persistent cryoglobulinemia after antiviral treatment.

Careful monitoring of patients with HCV infection who have liver cirrhosis and those at risk of other organ cancers is strongly recommended. Moreover, chronic HCV infection has been associated with an elevated risk of developing cancers in organs other than the liver, such as non-Hodgkin’s lymphoma and certain types of gastrointestinal cancers. Early detection and intervention strategies play a crucial role in improving patient prognosis and overall quality of life.

The early detection of HCV infection, prior to the progression of liver function deterioration, and subsequent eradication of the virus are paramount in low-income countries. Timely diagnosis allows for prompt initiation of treatment interventions, which can prevent or delay the onset of severe liver complications such as cirrhosis and HCC. Additionally, early treatment can minimize the socioeconomic burden associated with advanced liver disease, including the need for costly medical interventions and the loss of productivity due to illness. Furthermore, by focusing efforts on early detection and treatment, low-income countries can mitigate the long-term health and economic consequences of chronic HCV infection, ultimately improving the overall well-being of their populations. Therefore, prioritizing strategies for early detection and treatment of HCV infection is essential in resource-constrained settings to effectively combat the disease burden and reduce its societal impact.

The recurrence of serum cryoglobulinemia following antiviral medication increases the risk of complications from cryoglobulinemia-related skin, renal, joint, and neurological problems. As a result, re-evaluation of cryoglobulinemia in HCV-infected individuals, even after antiviral therapy, is critical to preventing future organ system problems.

Our study holds significant strengths, as it systematically examined cryoglobulinemia at the onset of treatment, revealing a noteworthy trend: patients initially testing negative subsequently tested positive following treatment completion. This observation emphasizes the critical need for ongoing monitoring of cryoglobulinemia among HCV-infected individuals at defined intervals. Such vigilance can effectively mitigate the risk of severe cryoglobulinemic manifestations and associated systemic diseases. Furthermore, our findings advocate for routine cryoglobulinemia screenings at the conclusion of antiviral therapy, regardless of the patient’s initial negative status. This proactive approach ensures comprehensive patient care and timely intervention to address potential complications. Our study has several limitations. Firstly, the number of patients with cryoglobulinemia included in our study was low, which may limit the generalizability of our findings. Second, we were unable to assess the clinical manifestations of MC, such as skin lesions, joint pain, and peripheral neuropathy, which are important factors in understanding the impact of cryoglobulinemia on CHC patients. Thirdly, we did not have the opportunity to include the duration of HCV infection before the initiation of DAA treatment, which may be related to the development of cryoglobulinemia. Additionally, it is crucial to study the clinical manifestations of MC before and after treatment to assess treatment efficacy and impact on patient outcome. 

Further studies are necessary to address these limitations and to provide a more comprehensive understanding of the relationship between HCV infection, antiviral treatment, and cryoglobulinemia. Future research directions may include larger sample sizes, comprehensive assessment of clinical manifestations, and longitudinal studies to assess long-term outcomes and treatment efficacy. It is essential for authors to discuss the results within the context of previous studies and working hypotheses, considering the broader implications and potential clinical relevance of the findings. 

## 5. Conclusions

In conclusion, 3 months after the conclusion of antiviral DAA therapy, eight patients tested positive for cryoglobulinemia, representing a prevalence of 6.7%. Liver cirrhosis and other organ cancers were independently and significantly associated with positive cryoglobulinemia 3 months after antiviral treatment. Further investigation into the causes of positive cryoglobulinemia after DAA antiviral therapy is warranted. These findings underscore the importance of continued monitoring and management of cryoglobulinemia in HCV-infected individuals, even after successful antiviral treatment, to prevent potential complications and improve patient outcomes.

## Figures and Tables

**Figure 1 diagnostics-14-01188-f001:**
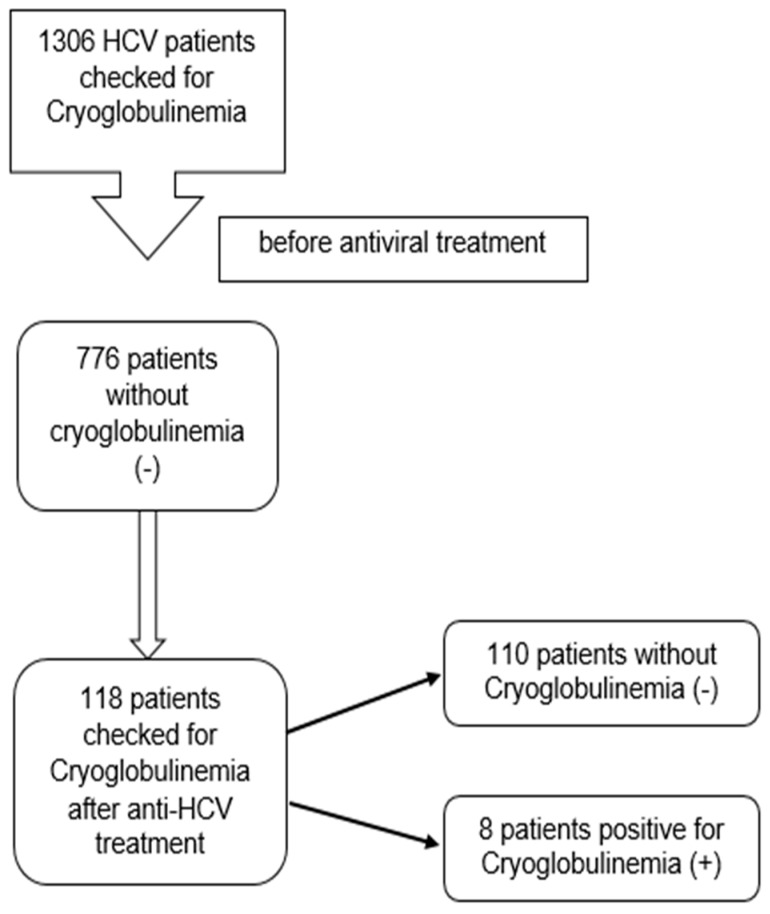
Study population.

**Figure 2 diagnostics-14-01188-f002:**
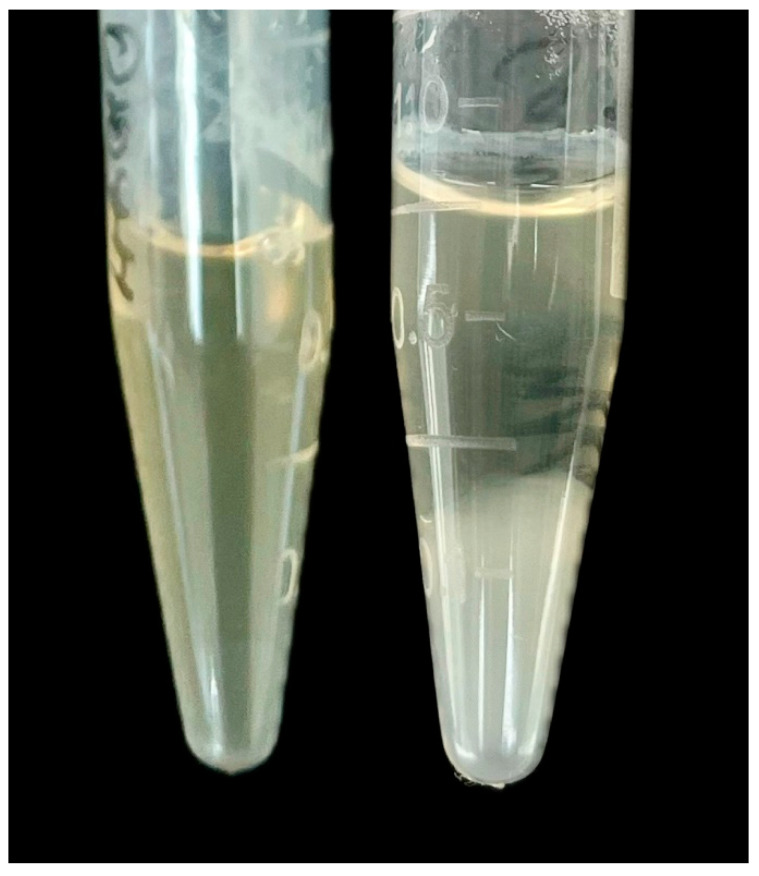
Detected cryoprecipitation after refrigeration (4 °C) for 7 days.

**Table 1 diagnostics-14-01188-t001:** Baseline characteristics compared in cryoglobulinemia positive and negative patients in 3 months after completion of antiviral treatment.

Characteristics	Total *n* = 118	Negative Cryoglobulinemia *n* = 110	Positive Cryoglobulinemia *n* = 8	*p* Value
Age (mean ± SD)	64.7 ± 11.5	64.2 ± 11.4	72.1 ± 10.4	0.061
Sex, *n* (%)				0.281
Male	63 (53.4%)	57 (51.8%)	6 (75%)	
Female	55 (46.6%)	53 (48.2%)	2 (25%)	
Body mass index (mean ± SD)	27.88 ± 9.3	27.6 ± 9.5	30.7 ± 5.0	0.365
HCV RNA (log IU/mL)	13.6 ± 2.1	13.6 ± 2.1	13.2 ± 1.6	0.635
AST (mean ± SD)	59.1 ± 55.0	58.5 ± 55.6	67.8 ± 46.9	0.668
ALT (mean ± SD)	62.6 ± 62.7	62.3 ± 63.6	67.4 ± 51.1	0.838
AFP (mean ± SD)	8.7 ± 36.2	5.3 ± 4.4	54.0 ± 135.4	0.343
Triglycerides (mean ± SD)	119.3 ± 68.0	119.3 ± 68.5	118.5 ± 66.2	0.976
Cholesterol (mean ± SD)	171.3 ± 37.6	171.3 ± 37.9	171.5 ± 35.9	0.989
Ferritin (mean ± SD)	320.8 ± 356.2	315.0 ± 334.3	406.8 ± 639.9	0.544
White blood cell (mean ± SD)	6.1 ± 1.8	6.1 ± 1.9	4.7 ± 1.3	0.058
Hemoglobin (mean ± SD)	12.9 ± 2.1	12.9 ± 2.1	13.3 ± 2.1	0.626
Platelet (mean ± SD)	191.3 ± 90.1	192.6 ± 91.9	170.8 ± 56.6	0.537
Creatinine (mean ± SD)	2.3 ± 2.8	2.4 ± 2.9	1.3 ± 0.3	**0.012**
FIB4 (mean ± SD)	3.0 ± 2.1	3.0 ± 2.0	4.1 ± 2.9	0.193
APRI (mean ± SD)	0.9 ± 0.9	0.8 ± 0.8	1.3 ± 1.4	0.448
CPT score (mean ± SD)	5.0 ± 0.3	5.0 ± 0.3	5.0 ± 0.0	0.478

Bold values are statistically significant. SD: standard deviation; AST: aspartate aminotransferase; ALT: alanine aminotransferase; AFP: alpha-fetoprotein; FIB4: fibrosis 4 index; APRI: AST to platelet ratio index; CPT: Child–Turcotte–Pugh score.

**Table 2 diagnostics-14-01188-t002:** Clinical characteristics compared in cryoglobulinemia positive and negative patients in 3 months after completion of antiviral treatment. Bold values are statistically significant.

Characteristics	Total *n* = 118	Negative Cryoglobulinemia *n* = 110	Positive Cryoglobulinemia *n* = 8	*p* Value
HBV coinfection, *n* (%)				0.851
Yes	11 (9.3%)	9 (8.2%)	2 (25%)	
No	107 (90.7%)	101 (91.8%)	6 (75%)	
HIV coinfection, *n* (%)				1.00
Yes	2 (1.8%)	2 (1.9%)	0 (0%)	
No	116 (98.2%)	108 (98.1%)	8 (100%)	
Liver cirrhosis, *n* (%)				**0.033**
Yes	21 (17.8%)	17 (15.5%)	4 (50%)	
No	97 (82.2%)	93 (84.5%)	4 (50%)	
Decompensated cirrhosis, *n* (%)				0.851
Yes	4 (3.4%)	4 (3.6%)	0 (0%)	
No	114 (96.6%)	106 (96.4%)	8 (100%)	
Hepatocellular carcinoma				0.588
Yes	12 (10.2%)	11 (10.0%)	1 (12.5%)	
No	106 (89.8%)	99 (90.0%)	7 (87.5%)	
Diabetes, *n* (%)				0.851
Yes	4 (3.4%)	4 (3.6%)	0 (0%)	
No	114 (96.6%)	106 (96.4%)	8 (100%)	
Hypertension, *n* (%)				0.321
Yes	2 (1.8%)	2 (1.9%)	0 (0%)	
No	116 (98.2%)	108 (98.1%)	8 (100%)	
Hyperlipidemia, *n* (%)				0.809
Yes	5 (4.5%)	5 (4.6%)	0 (0%)	
No	113 (95.5%)	105 (95.4%)	8 (100%)	
Cerebrovascular event, *n* (%)				1.00
Yes	7 (5.9%)	7 (6.4%)	0 (0%)	
No	111 (94.1%)	103 (93.2%)	8 (100%)	
Cardiovascular event, *n* (%)				0.954
Yes	14 (11.9%)	13 (11.8%)	1 (12.5%)	
No	104 (88.1%)	97 (88.2%)	7 (87.5%)	
Chronic kidney disease, *n* (%)				0.232
Yes	7 (5.9%)	7 (6.4%)	0 (0%)	
No	111 (94.1%)	103 (93.2%)	8 (100%)	
Cancer other organ, *n* (%)				**0.006**
Yes	22 (18.6%)	17 (15.5%)	5 (62.5%)	
No	96 (81.4%)	93 (84.5%)	3 (375%)	

**Table 3 diagnostics-14-01188-t003:** Multivariate analysis of the factors associated with positivity of cryoglobulinemia 3 months after antiviral treatment. Bold values are statistically significant.

Characteristics	Univariate Analysis		Multivariate Analysis	
	OR (95% CI)	*p* Value	OR (95% CI)	*p* Value
Cirrhosis	5.47 (1.24–24.01)	**0.024**	17.86 (1.79–177.35)	**0.014**
Other organ cancer	9.11 (1.99–41.76)	**0.004**	25.17 (2.59–244.23)	**0.005**
Age (>70)	2.23 (0.52–9.46)	0.275	1.23 (0.22–6.71)	0.806

## Data Availability

Data are contained within the article.

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
