# Peer review of "Post-Treatment Occurrence of Serum Cryoglobulinemia in Chronic Hepatitis C Patients"

_diagnostics, 2024, doi:10.3390/diagnostics14111188_

Round 1
Reviewer 1 Report
Comments and Suggestions for Authors
In this study, Dashjamts et al. examine cryoglobulinemia in HCV patients after receiving antiviral therapy. Overall, the study lacks the number of patients and other methods to confirm the manifestation of cryoglobulinemia. I also have other comments, as described below.
1. Why do only 118 out of 776 patients undergo testing for cryoglobulinemia after the anti-HCV treatment? What’s happened with the rest of the cohort? Given that 1306 patients had already undergone cryoglobulinemia testing prior to the anti-HCV treatment, the authors should have conducted additional tests to confirm the cryoglobulinemia finding.
2. Do patients who tested positive for cryoglobulinemia before the anti-HCV treatment retain this clinical manifestation after the antiviral therapy?
3. The authors mention in the introduction that “clinical remission of MC syndrome in chronic infection with HCV is often associated with clearance of the virus.” (line 70) But in this study, the authors suggest that the antiviral treatment promotes cryoglobulinemia. How do the authors explain this disagreement? Doesn't the antiviral treatment lead to the virus's clearance?
4. In light of the previous comments, what is the status of HCV after the anti-HCV treatment?
5. The authors should provide a discussion of potential molecular mechanisms by which the antiviral treatment promotes cryoglobulinemia.
Comments on the Quality of English Language
There are several grammatical errors throughout the text. For the revision, the authors should proofread carefully.
Author Response
Reviewers' comments:
Reviewer #1: General comments
In this study, Dashjamts et al. examine cryoglobulinemia in HCV patients after receiving antiviral therapy. Overall, the study lacks the number of patients and other methods to confirm the manifestation of cryoglobulinemia. I also have other comments, as described below.
Specific comments
- Why do only 118 out of 776 patients undergo testing for cryoglobulinemia after the anti-HCV treatment? What’s happened with the rest of the cohort? Given that 1306 patients had already undergone cryoglobulinemia testing prior to the anti-HCV treatment, the authors should have conducted additional tests to confirm the cryoglobulinemia finding.
Thank you very much for this comment. We have checked cryogolubulinemia almost one seventh of patients with positive for cryoglobulinemia after antiviral treatment. However, we addressed this issue in the limitation section of discussion. Also all statistical analysis have been made in 118 patients with chronic hepatitis C patients after antiviral treatment.
- Do patients who tested positive for cryoglobulinemia before the anti-HCV treatment retain this clinical manifestation after the antiviral therapy?
Thank you very much for this comment. If cryoglobulinemia is not diagnosed during antiviral therapy, the clinical symptoms and consequences may improve. If cryoglobulinemia continues, other diagnostic and treatment options should be considered.
- The authors mention in the introduction that “clinical remission of MC syndrome in chronic infection with HCV is often associated with clearance of the virus.” (line 70) But in this study, the authors suggest that the antiviral treatment promotes cryoglobulinemia. How do the authors explain this disagreement? Doesn't the antiviral treatment lead to the virus's clearance?
Thank you very much for this reminder. In our next manuscript this issue must be investigated. Thank you again. There are rare studies which were investigated about spontaneous clearance of cryoglobulinemia in hepatitis C virus infected individuals. Unfortunately, we do not have untreated hepatitis C patient’s data. Spontaneous clearance of cryoglobulinemia is maybe not possible unless we do the treatment which is related to plasmapheresis or therapeutic plasma exchange.
- In light of the previous comments, what is the status of HCV after the anti-HCV treatment?
A sustained virologic response is associated with lower all-cause mortality and improves hepatic and extrahepatic manifestations, cognitive function, physical health, work productivity, and quality of life. Having SOF and other DAAs will definitely benefit Mongolian HCV patients, but efforts should be made to make DAAs accessible to all patients.
- The authors should provide a discussion of potential molecular mechanisms by which the antiviral treatment promotes cryoglobulinemia.
Thank you very much for this reminder. We have added a paragraph in discussion section.
Reviewer 2 Report
Comments and Suggestions for Authors
It is a study about the possibility of cryoglobulinemia presence in HCV patients after HCV eradication.
I have some significant comments to make
1. The authors reported that they screened 1306 patients for cryoglobulinemia, and 776 were found to be negative. Later, they decided to investigate 118 of these patients after HCV treatment. It is vital to provide a detailed description of how the authors selected and included these particular 118 patients to avoid any potential bias.
2. The authors need to provide more details regarding the excluded patients and the specific reasons for their exclusion.
3. The sentence "categorical variables compared using a t-test" is incorrect. Please correct it.
4. Line 218. The statement "Individuals with cryoglobulinemia have a decreased risk of developing hepatocellular carcinoma, diabetes, hypertension, hyperlipidemia, chronic kidney disease, cardiovascular disease, and cerebrovascular disease compared to those without cryoglobulinemia" is incorrect. While patients with cryoglobulinemia did have lower rates of these conditions, they did not have a decreased risk of developing them. Please note the distinction.
5. What results led the authors to include age>70 in the univariate and multivariate analyses?
6. During the discussion, the authors stated that a lower prevalence rate of cryoglobulinemia was found in patients with co-infection of HCV/HBV and HCV/HIV in this study. However, this conclusion is misleading since the number of patients in these groups was small, and we cannot draw any definitive conclusion from this data.
7. The authors should remove the paragraph about the possible correlation between RF or serum lipids and cryoglobulinemia, as it is beyond the scope of this study.
8. The authors must clarify the new information this study brings to the existing literature and how it can be applied in clinical practice. They should also emphasize the significance of their findings. Additionally, as the number of patients with cryoglobulinemia is limited, the authors should avoid discussing irrelevant information and instead concentrate on the study's potential strengths.
Comments on the Quality of English LanguageThe text needs editing in English because it contains several errors in grammar and syntax.
Author Response
Reviewers' comments:
Reviewer #2: General comments
It is a study about the possibility of cryoglobulinemia presence in HCV patients after HCV eradication.
I have some significant comments to make
- The authors reported that they screened 1306 patients for cryoglobulinemia, and 776 were found to be negative. Later, they decided to investigate 118 of these patients after HCV treatment. It is vital to provide a detailed description of how the authors selected and included these particular 118 patients to avoid any potential bias.
Thank you very much for this comment. There were 118 patients who could be tested for cryoglobulinemia in their blood following antiviral therapy. Also all statistical analysis have been made in 118 patients with chronic hepatitis C patients after antiviral treatment. However, we addressed this issue in the limitation section of discussion.
- The authors need to provide more details regarding the excluded patients and the specific reasons for their exclusion.
Thank you very much for this reminder and we have added this information in Materials and Methods section.
- The sentence "categorical variables compared using a t-test" is incorrect. Please correct it.
Thank you very much for this reminder and we have corrected it.
- Line 218. The statement "Individuals with cryoglobulinemia have a decreased risk of developing hepatocellular carcinoma, diabetes, hypertension, hyperlipidemia, chronic kidney disease, cardiovascular disease, and cerebrovascular disease compared to those without cryoglobulinemia" is incorrect. While patients with cryoglobulinemia did have lower rates of these conditions, they did not have a decreased risk of developing them. Please note the distinction.
Thank you very much for this reminder and we have corrected it.
- What results led the authors to include age>70 in the univariate and multivariate analyses?
Thank you very much for this comment. We have checked for the association between persistent cryoglobulinemia and age>70. Following antiviral medication, elderly people had a greater frequency of cryoglobulinemia. The extended circulation of cryoglobulinemia precipitates in the serum might be the cause of this. This suggests that among virally infected people, we should investigate the correlation between age and cryoglobulinemia.
- During the discussion, the authors stated that a lower prevalence rate of cryoglobulinemia was found in patients with co-infection of HCV/HBV and HCV/HIV in this study. However, this conclusion is misleading since the number of patients in these groups was small, and we cannot draw any definitive conclusion from this data.
Thank you very much for this comment. In our study, HCV/HBV and HCV/HIV co-infected subjects were less likely to develop cryoglobulinemia after antiviral therapy. It should detect cryoglobulinemia after antiviral therapy in more people with more co-infections.
- The authors should remove the paragraph about the possible correlation between RF or serum lipids and cryoglobulinemia, as it is beyond the scope of this study.
Thank you very much for this reminder and we have corrected it.
- The authors must clarify the new information this study brings to the existing literature and how it can be applied in clinical practice. They should also emphasize the significance of their findings. Additionally, as the number of patients with cryoglobulinemia is limited, the authors should avoid discussing irrelevant information and instead concentrate on the study's potential strengths.
Thank you very much for this reminder and we have added this information in the Discussion section.
Round 2
Reviewer 1 Report
Comments and Suggestions for Authors
I thank the reviewer for addressing my questions and concerns. I understand the limitations of the current study. This work could proceed for publication.
Comments on the Quality of English LanguageOverall, it is fine.
Author Response
I thank the reviewer for addressing my questions and concerns. I understand the limitations of the current study. This work could proceed for publication.
Thank you for your insightful review.
Reviewer 2 Report
Comments and Suggestions for Authors
The revised version of the manuscript is better than the old version. Grammar and syntax errors have sufficiently been corrected. However, the authors did not adequately answer three of my previous comments.
1. Regarding the first comment, my concern was not the number of included patients, i.e., whether this was large enough. The authors yielded 118 patients who had been treated with anti-HCV therapy. This is clear. However, the authors need to clarify how these patients were chosen. Do these 118 patients comprise the whole group of patients that managed to be treated among the total sum of 776 patients? Or were these patients randomly selected for treatment among other candidates from the total group of 776 patients? Or do they comprise a group of 118 consecutive patients that the authors had decided to put on treatment in a pre-specified study period?
If you do not adequately explain, we cannot be sure about any risk of bias.
2. Line 222. The term "risk" needs to be revised. We use the term "risk" when something predisposes to the development of something else, for example, if cryoglobulinemia leads to the development of comorbidities such as chronic kidney disease, cardiovascular disease, cerebrovascular disease, etc. The authors did not investigate this issue. They simply noted whether patients with cryoglobulinemia had different proportions of the above comorbidities than those without. They did not explore whether cryoglobulinemia was a predictor independently associated with an increased or decreased risk of developing comorbidities. It would be more accurate to state whether the presence of these comorbidities differed or not between the two groups rather than referring to the "risk."
3. The authors should explain the study's strengths and contributions to the existing literature more in-depth and elaborate on how the study results could impact clinical practice.
Author Response
Reviewer #2: General comments
The revised version of the manuscript is better than the old version. Grammar and syntax errors have sufficiently been corrected. However, the authors did not adequately answer three of my previous comments.
- Regarding the first comment, my concern was not the number of included patients, i.e., whether this was large enough. The authors yielded 118 patients who had been treated with anti-HCV therapy. This is clear. However, the authors need to clarify how these patients were chosen. Do these 118 patients comprise the whole group of patients that managed to be treated among the total sum of 776 patients? Or were these patients randomly selected for treatment among other candidates from the total group of 776 patients? Or do they comprise a group of 118 consecutive patients that the authors had decided to put on treatment in a pre-specified study period?
If you do not adequately explain, we cannot be sure about any risk of bias.
Thank you very much for this insightful comment. These 118 patients were able to check cryoglobulinemia after 3 months of cessation of antiviral treatment and also they were randomly selected from 776 patients. We have added this explanation in Patients section on Materials and Methods.
- Line 222. The term "risk" needs to be revised. We use the term "risk" when something predisposes to the development of something else, for example, if cryoglobulinemia leads to the development of comorbidities such as chronic kidney disease, cardiovascular disease, cerebrovascular disease, etc. The authors did not investigate this issue. They simply noted whether patients with cryoglobulinemia had different proportions of the above comorbidities than those without. They did not explore whether cryoglobulinemia was a predictor independently associated with an increased or decreased risk of developing comorbidities. It would be more accurate to state whether the presence of these comorbidities differed or not between the two groups rather than referring to the "risk."
Thank you very much for this reminder and we have changed “risk” to “characteristics” in line 225.
- The authors should explain the study's strengths and contributions to the existing literature more in-depth and elaborate on how the study results could impact clinical practice.
Thank you very much for this comment and we have added this information in the Discussion section.
Round 3
Reviewer 2 Report
Comments and Suggestions for Authors
I have nothing more to mention. The authors adequately answered my questions.
Comments on the Quality of English LanguageThe quality of English Language is fine